# Statistical population reconstruction of moose (*Alces alces*) in northeastern Minnesota using integrated population models

**William J. Severud**[1]☉*, **Sergey S. Berg**[2]☉, **Connor A. Ernst**[3], **Glenn D. DelGiudice**[4], **Seth A. Moore**[5], **Steve K. Windels**[6], **Ron A. Moen**[7], **Edmund J. Isaac**[5], **Tiffany M. Wolf**[1]

**1** Department of Veterinary Population Medicine, University of Minnesota, Saint Paul, Minnesota, United States of America, **2** Department of Computer and Information Sciences, University of St. Thomas, Saint Paul, Minnesota, United States of America, **3** Department of Mathematics, University of St. Thomas, Saint Paul, Minnesota, United States of America, **4** Forest Wildlife Populations and Research Group, Minnesota Department of Natural Resources, Forest Lake, Minnesota, United States of America, **5** Department of Biology and Environment, Grand Portage Band of Lake Superior Chippewa, Grand Portage, Minnesota, United States of America, **6** Voyageurs National Park, International Falls, Minnesota, United States of America, **7** Center for Water and the Environment, University of Minnesota, Duluth, Minnesota, United States of America

☉ These authors contributed equally to this work.
* william.severud@sdstate.edu

**Data Availability Statement:** All relevant data are within the paper in Tables 1, 2, and 3.

## Abstract

Given recent and abrupt declines in the abundance of moose (*Alces alces*) throughout parts of Minnesota and elsewhere in North America, accurately estimating statewide population trends and demographic parameters is a high priority for their continued management and conservation. Statistical population reconstruction using integrated population models provides a flexible framework for combining information from multiple studies to produce robust estimates of population abundance, recruitment, and survival. We used this framework to combine aerial survey data and survival data from telemetry studies to recreate trends and demographics of moose in northeastern Minnesota, USA, from 2005 to 2020. Statistical population reconstruction confirmed the sharp decline in abundance from an estimated 7,841 (90% CI = 6,702–8,933) in 2009 to 3,386 (90% CI = 2,681–4,243) animals in 2013, but also indicated that abundance has remained relatively stable since then, except for a slight decline to 3,163 (90% CI = 2,403–3,718) in 2020. Subsequent stochastic projection of the population from 2021 to 2030 suggests that this modest decline will continue for the next 10 years. Both annual adult survival and per-capita recruitment (number of calves that survived to 1 year per adult female alive during the previous year) decreased substantially in years 2005 and 2019, from 0.902 (SE = 0.043) to 0.689 (SE = 0.061) and from 0.386 (SE = 0.030) to 0.303 (SE = 0.051), respectively. Sensitivity analysis revealed that moose abundance was more sensitive to fluctuations in adult survival than recruitment; thus, we conclude that the steep decline in 2013 was driven primarily by decreasing adult survival. Our analysis demonstrates the potential utility of using statistical population reconstruction to monitor moose population trends and to identify population declines more quickly. Future studies should focus on providing better estimates of per-capita recruitment, using

**Funding:** CAE was financially supported in part by the Center for Applied Mathematics Summer Research Program at the University of Saint Thomas (https://cas.stthomas.edu/departments/areas-of-study/mathematics/center-for-applied-mathematics/). Moose collaring work was funded by Voyageurs National Park (https://www.nps.gov/voya/index.htm; SKW and RAM), the USGS-NPS Natural Resource Preservation Program (PMIS 140435; SKW and RAM), University of Minnesota-Duluth (https://nrri.umn.edu/; SKW and RAM), U.S. Fish and Wildlife Service Tribal Wildlife Grant (https://www.fws.gov/service/tribal-wildlife-grants; SAM), U.S. Environmental Protection Agency Great Lakes Restoration Initiative (https://www.glri.us/node/443, SAM), and the Bureau of Indian Affairs Endangered Species Program (https://www.bia.gov/bia/ots/division-natural-resources/branch-fish-wildlife-recreation/endangered-species-program; SAM), Minnesota Zoo Ulysses S. Seal Conservation Fund (https://mnzoo.org/conservation/around-world/ulysses-s-seal-conservation-grant-program/; TMW), Indianapolis Zoo Conservation Fund (https://www.indianapoliszoo.com/conservation/field-support/; TMW). The funders had no role in study design, data collection and analysis, decision to publish, or preparation of the manuscript.

**Competing interests:** The authors have declared that no competing interests exist.

pregnancy rates and calf survival, which can then be incorporated into reconstruction models to help improve estimates of population change through time.

## Introduction

Effective management and conservation of wildlife species requires an accurate understanding of population abundance, recruitment, survival, and age- and sex-ratios, and how these parameters change over time and in response to various extrinsic factors, such as hunting and habitat alteration. Unfortunately, accurately estimating abundance and demographic parameters is challenging, because direct monitoring of animals is often costly and impractical, particularly in densely forested regions or for animals that occur at low densities. Given these difficulties, most abundance estimates have relied on methods that are limited to small geographical areas or sample sizes, including track surveys [1], analysis of camera traps [2], and telemetry data [3]. Each of these methods by themselves do not provide a cost-effective means of estimating abundance and other demographic parameters across larger spatial scales at which most management occurs.

Statistical population reconstruction using integrated population models (IPMs) has emerged as a flexible framework for combining information from multiple studies using various, disparate datasets (e.g., aerial surveys, radio-collared individuals, age-at-harvest), and even from different parts of a state or region, to provide a more robust and cost-effective means of estimating species abundance and demographics across large spatial scales [4, 5]. This method simultaneously estimates multiple demographic parameters (e.g., annual abundance, recruitment, and survival) and their uncertainties throughout time, and can be used to provide separate estimates for different sexes and age classes. Such models have previously been used to estimate abundance and trends of wildlife species, such as American marten (*Martes americana*), black bears (*Ursus americanus*), and mountain lions (*Puma concolor*) [6–8].

Accurately estimating the abundance and trajectory of the moose (*Alces alces*) population in northeastern Minnesota (MN) is of current interest due to a recent and abrupt decline that was detected via aerial surveys between 2010 and 2013 [9]. At its nadir in 2013, this population estimate was 69% lower than when at its peak in 2006 (2,760 versus 8,840), but it appeared to have stabilized during 2012–2020 as estimated by aerial surveys [9, 10]. A study of demographics of the northeastern population in 2002–2008 predicted a slow reduction in numbers (long-term stochastic annual growth rate [λ] of 0.85,) with modeled adult and calf survival rates of 0.74–0.85 and 0.24–0.56, respectively [11]. However, the abrupt decline in northeastern MN was not detected by the annual aerial surveys until 2010 [11–13], which illustrated that demographic modeling may reveal population trajectories before they are reflected in total population estimates by aerial survey.

In response to the rapid decline of moose in the northeastern population, the MN Department of Natural Resources (MNDNR), Grand Portage Band of Lake Superior Chippewa, and Voyageurs National Park all independently initiated studies of adult and calf survival and cause-specific mortality (Fig 1). These studies built upon previous research [11, 14], but aimed to better understand causes of mortality [15, 16]. The more recent research employed global positioning system (GPS) collars and other remote monitoring techniques (e.g., internal temperature monitors, movement analyses) to track survival, habitat use, causes of mortality, physiological condition, and disease transmission dynamics [17–27].

Our goal was to integrate these multiple data streams into a unified model that would accurately describe past population dynamics and future projections of the northeastern MN moose population. Specifically, we used statistical population reconstruction to estimate population abundance, recruitment, and survival rates using all available data. We also examined the sensitivity of model estimates to fluctuations in adult survival and per-capita recruitment (number of calves that survived to 1 year per adult female alive during the previous year) to determine which may be more important in predicting population growth and used time series analysis to project population estimates 10 years into the future to inform management and conservation concerns. Given recent declines in moose abundance occurring broadly across North America [28, 29], our study demonstrates the utility of statistical population reconstruction for understanding moose population dynamics.

## Materials and methods

### Study area

Our study occurred in northeastern MN, near the southern limit of the distributional range of moose (Fig 1) [11, 28]. Our study area was a mosaic of the Superior National Forest and various Tribal, state, county, and private lands (Fig 1), as well as the federal lands of Voyageurs National Park (VNP). Moose are a subsistence food used by the Anishinaabeg (people) of the Grand Portage Band of Lake Superior Chippewa historically and presently. The Grand Portage Band is a federally recognized Indian tribe in extreme northeastern MN and proudly exercises its rights to food sovereignty through subsistence hunting and fishing. Voyageurs National Park is just west of primary moose range, which is delineated by MNDNR Section of Wildlife field and research staff (Fig 1). Moose occur outside of primary range, but at low densities. Statewide moose harvest was closed during 1922–1971, because of low moose numbers, and then reopened in the northwestern and northeastern portions of the state with limited permits issued [30]. Harvest was stopped in the northwest in 1997, but continued in the northeast. In 2007, hunters were restricted to harvesting antlered adult males only [30]. Moose harvests were then suspended in MN from 2013 until 2016, when a tribal subsistence harvest was resumed [31–33]. Moose harvests do not occur in VNP.

Our study area is part of the Northern Superior Upland within the Laurentian mixed forest province [34]. The vegetative cover is a mosaic of wetlands, stands of northern white cedar (*Thuja occidentalis*), black spruce (*Picea mariana*), tamarack (*Larix laricina*), and upland stands of balsam fir (*Abies balsamea*), jack pine (*Pinus banksiana*), eastern white pine (*P. strobus*), and red pine (*P. resinosa*), intermixed with quaking aspen (*Populus tremuloides*) and paper birch (*Betula papyrifera*).

Moose range in this region overlapped with gray wolves (*Canis lupus*) and American black bears, both of which prey upon adult and calf moose [14, 20, 25, 35, 36]. Adult and calf moose hair was present in relatively few wolf scats from VNP (0–4% occurrence) compared to scats from other areas of moose range in MN (7–22% occurrence) [37, 38]. The moose population in northeastern MN were afflicted by various parasites and disease, including infestation by winter ticks (*Dermacentor albipictus*) and infection by meningeal worm (*Parelaphostrongylus tenuis*) and giant liver fluke (*Fascioloides magna*) [24, 26, 39].

### Aerial surveys

As part of the ongoing monitoring and management of moose in northern MN that have taken place since the 1960s, the MNDNR, in cooperation with Fond du Lac Band of Lake Superior Chippewa (FDL) and 1854 Treaty Authority, conducted an aerial survey of the northeastern moose population each winter using an updated and standardized approach since 2005

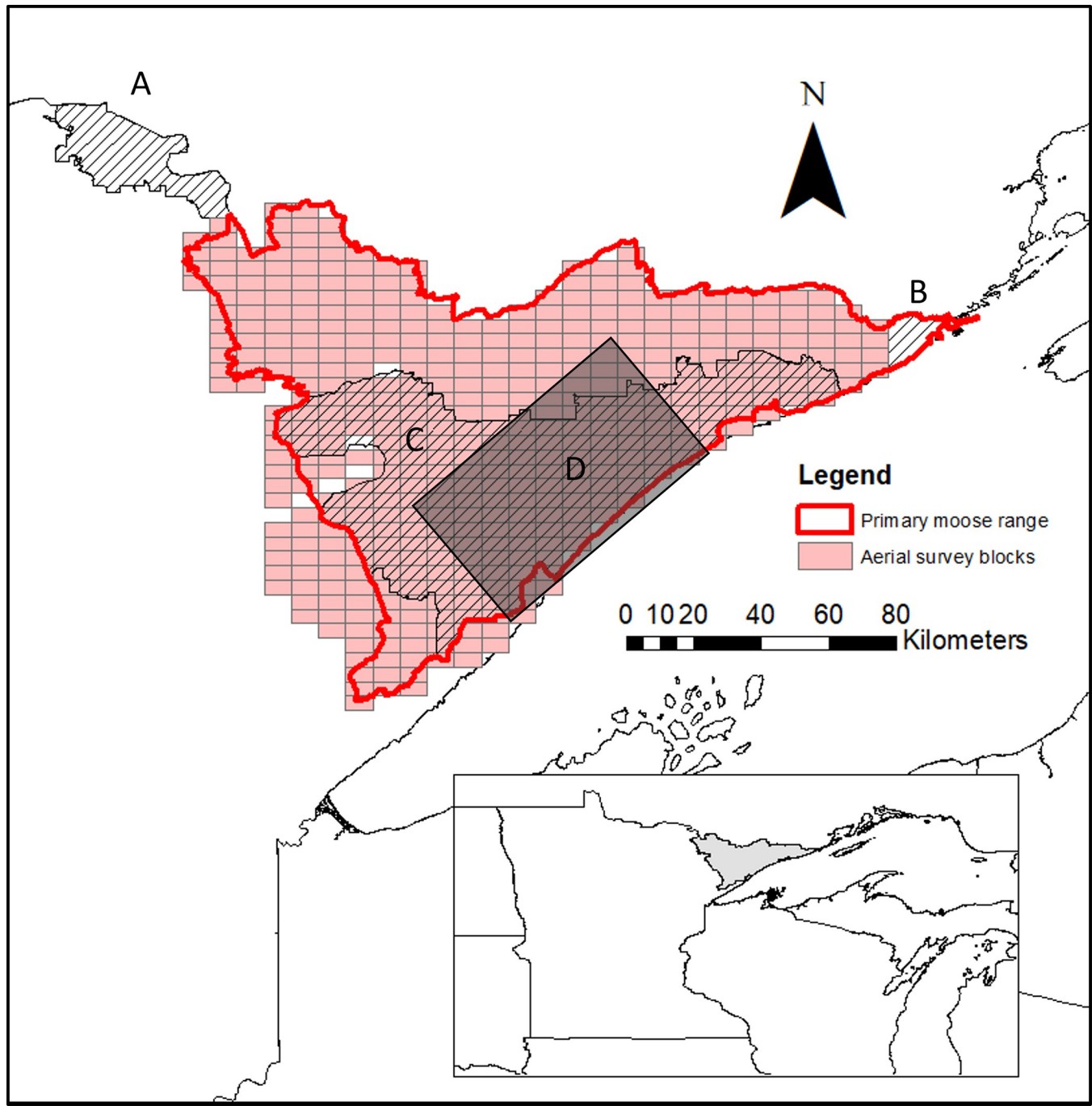

**Fig 1. Study area map.** Primary moose range in Minnesota (red outline) that is surveyed annually by cooperators Minnesota Department of Natural Resources (MNDNR), Fond du Lac Band of Lake Superior Chippewa (FDL), and 1854 Treaty Authority; and 4 study areas that contained collared moose: Voyageurs National Park (A), Grand Portage Indian Reservation (B), MNDNR study (2012–2016; C), and MNDNR-FDL-1854 Treaty Authority study (2005–2008; D).

[9]; however, a survey was not conducted in 2021 due to the COVID-19 pandemic. Timing of surveys was typically during the first two weeks of January; however, insufficient snow depth postponed the 2012 survey until 26 January to 9 February [40]. The surveys were conducted using helicopters over a total area of approximately 15,500 km$^2$. This area was divided into 436

rectangular survey plots of approximately 36 km$^2$ each, 36 to 52 of which were selected each year using a stratified random sampling protocol based on moose density (low, medium, high). Moose density strata were classified collaboratively by MNDNR, FDL, and 1854 Treaty Authority staff and are reevaluated every 5 years based on expert knowledge and previous survey results. Each sighted moose was classified as either a calf, adult female, or adult male based on body size and presence of vulva patch and/or antlers; uncorrected estimates (without a sightability correction) adjusted for sampling were then used to calculate adult male:female and calf:adult female ratios at the population level [9, 41]. A sightability model was then used to estimate overall abundance. Visual obstruction was calculated as the proportion of area within a 10-m radius surrounding the first moose observed in a group that was not visible and used to adjust each estimate and corresponding 90% confidence intervals (CI; Table 1) [9, 41]. We used the estimated annual abundance of calves, adult females, and adult males derived from the aerial surveys in the IPM below. We scaled the variance of the overall point count on the proportion of calves to obtain variance estimates for calf abundance.

## Adult survival rates

In addition to aerial survey data, we used adult moose survival data collected via telemetry from 2005 to 2019 by four different studies throughout northeastern MN (Fig 1). We excluded animals with collar failures from the data in the year of collar failure (i.e., right-censoring), animals that died as a result of capture, and young-of-the-year from any further analysis. Collar failure was assumed to be independent of moose fate. The remaining animals in each study were pooled together to determine annual mortality and associated at-risk counts as a measure of adult survival rates (Table 2).

We used annual adult survival rates from two previous studies by MNDNR, FDL, and the 1854 Treaty Authority [11, 14, 24, 42]. The earlier MNDNR-FDL-1854 Treaty Authority study used 150 adult moose (95 F/55 M) collared during 2002–2008 [11]; however, we only used survival rates that coincided with the aerial survey (2005–2007). We used pooled adult survival

**Table 1. Moose population estimates by year, sex, and age class.** Age-class-specific aerial survey data with corresponding annual totals and 90% confidence intervals for moose in northeastern Minnesota, USA, 2005–2020 [9]. Total abundance is corrected for sightability, abundance of calves, adult females, and adult males is derived from reported calf:adult female and adult male:adult female ratios.

| Year | Calf | Adult female | Adult male | Total |
|------|------|--------------|------------|-------|
| 2005 | 1,658 | 3,188 | 3,315 | 8,160 (6,090–11,410) |
| 2006 | 1,237 | 3,638 | 3,965 | 8,840 (6,790–11,910) |
| 2007 | 913 | 3,147 | 2,801 | 6,860 (5,320–9,100) |
| 2008 | 1,334 | 3,704 | 2,852 | 7,890 (6,080–10,600) |
| 2009 | 1,110 | 3,469 | 3,261 | 7,840 (6,260–10,040) |
| 2010 | 756 | 2,701 | 2,242 | 5,700 (4,540–7,350) |
| 2011 | 626 | 2,606 | 1,668 | 4,900 (3,870–6,380) |
| 2012 | 624 | 1,734 | 1,872 | 4,230 (3,250–5,710) |
| 2013 | 356 | 1,078 | 1,326 | 2,760 (2,160–3,650) |
| 2014 | 714 | 1,623 | 2,013 | 4,350 (3,220–6,210) |
| 2015 | 439 | 1,513 | 1,498 | 3,450 (2,610–4,770) |
| 2016 | 689 | 1,641 | 1,690 | 4,020 (3,230–5,180) |
| 2017 | 588 | 1,634 | 1,487 | 3,710 (3,010–4,710) |
| 2018 | 428 | 1,157 | 1,446 | 3,030 (2,320–4,140) |
| 2019 | 539 | 1,633 | 2,008 | 4,180 (3,250–5,580) |
| 2020 | 502 | 1,394 | 1,254 | 3,150 (2,400–4,320) |

**Table 2. Number of moose that died and were at-risk by year and study.** Telemetry data from four different studies of annual mortality ($v$) and associated at-risk counts ($n$) for yearling and adult moose in northeastern Minnesota, USA, 2005–2019.

| Year | Lenarz et al. 2009 | | Carstensen et al. 2018 | | Voyageurs National Park | | Grand Portage Indian Reservation | |
|---|---|---|---|---|---|---|---|---|
| | $v$ | $n$ | $v$ | $n$ | $v$ | $n$ | $v$ | $n$ |
| 2005 | 13 | 51 | | | | | | |
| 2006 | 10 | 32 | | | | | | |
| 2007 | 10 | 57 | | | | | | |
| 2008 | | | | | | | | |
| 2009 | | | | | | | | |
| 2010 | | | | | 0 | 11 | 2 | 10 |
| 2011 | | | | | 2 | 19 | 5 | 15 |
| 2012 | | | | | 3 | 19 | 0 | 12 |
| 2013 | | | 20 | 105 | 0 | 14 | 9 | 22 |
| 2014 | | | 12 | 101 | 1 | 14 | 4 | 28 |
| 2015 | | | 14 | 93 | 2 | 11 | 8 | 38 |
| 2016 | | | 8 | 57 | 1 | 5 | 3 | 36 |
| 2017 | | | | | 1 | 4 | 4 | 31 |
| 2018 | | | | | | | 4 | 28 |
| 2019 | | | | | | | 2 | 29 |

estimates, because there was no difference in survival between males and females [11, 14]. The more recent MNDNR study was conducted from 2013 to 2016 and used 173 adult moose (123 F/50 M) [24]. Differences in survival between males and females were not reported, so we used the pooled adult survival estimates from this study. Details of animal capture, handling, collaring, and monitoring can be found in the source publications [11, 14, 24, 42].

We used 2 additional sources of adult moose survival data from study sites that are adjacent to the aerial survey area (Fig 1). Voyageurs National Park collared 21 moose (14 F/7 M) to study moose survival from 2010 to 2017. Grand Portage Indian Reservation collared 99 adult moose (76 F/23M) between 2010 and 2019. All capture and handling protocols were conducted in accordance with requirements of the University of MN Institutional Animal Care and Use Committee (protocols 1803-35736A and 0192A75532) and the guidelines of the American Society of Mammalogists [25, 43, 44]. We calculated Kaplan-Meier survival estimates using the "survival" package in Program R [45, 46]. Because adult moose captures typically occurred in mid-winter (Jan–Mar), we modeled annual survival using the calendar year (i.e., $t_0$ = 1 Jan) [47]. Collared moose that survived multiple years contributed an observation for each year they were alive, yielding 98 moose-years for Voyageurs National Park and 302 moose-years for Grand Portage. We used the "survdiff" function in the "survival" R package, which uses a log-rank test, to examine differences in overall survival between sexes [45, 46].

## Population reconstruction of moose in MN

Population reconstruction typically begins by specifying a projection matrix to describe the change in the number of animals in each cohort over time. Consider a hypothetical population of moose divided into four classes (male and female, calves and adults) monitored over $Y$ consecutive years, where $N_{ij}$ is the abundance in winter of animals of class $j$ in year $i$. Under this framework, all individuals born during the same year constitute a single cohort that is subsequently subjected to annual mortality from various causes. Previous reconstructions have then used an age-at-harvest matrix to represent each cohort [48–50]; however, with the exception of

tribal subsistence harvest averaging about 40 moose per year [32, 33], moose are not regularly harvested in MN. As such, we did not explicitly model the impacts of harvest mortality. In lieu of these data, we used aerial survey data to represent each cohort as a separate diagonal, where the observed counts, $a_{ij}$, are a function of the initial abundance of the corresponding cohort and the annual survival rate (to be estimated as parameters). Simulation studies have demonstrated that statistical reconstruction provides an unbiased estimate of population abundance [49]. Due to the difficulty associated with identifying sex of moose calves during aerial surveys, and the assumptions of a 50:50 sex-ratio of calves at birth with no sex differences in first year survival, we pooled male and female calves into a single cohort, for a total of $A = 3$ classes (calves, adult females, and adult males; Table 1). We defined adults as moose >1.5 years old, as they are classified in the aerial survey.

An objective function or estimator was then used to determine which set of model parameters best describes the observed data. We used a chi-square objective function to model the difference between the observed and predicted number of animals in each cohort and the joint difference for the entire matrix as

$$\Lambda_{Joint} = \sum_{i=1}^{Y} \sum_{j=1}^{A} \chi_{ij}^2,$$

where $\chi_{ij}^2$ is the cell-specific chi-square calculation [7, 51]. The difference for the cell represented by the total number of adult females in year 2 (i.e., $N_{22}$), for example, can be written as follows:

$$\chi_{22}^2 = \frac{(a_{22} - N_{22})^2}{N_{22}} = \frac{(h_{13} - (N_{11} \times 0.5 + N_{12}) \times S_1)^2}{(N_{11} \times 0.5 + N_{12}) \times S_1},$$

where $a_{22}$ is the number of adult females in year 1 observed via aerial survey, $N_{11}$ and $N_{12}$ are the initial calf and adult female cohort abundance in year 1, $S_1$ is the annual survival rate in year 1 (which we assumed to be constant for males and females but different between years), and 0.5 represents the sex-at-birth ratio to separate calves into adult females and males after the first year of life [52].

In addition to aerial survey data, we used information from collared individuals with known fates to help estimate annual survival by comparing the observed number of mortalities each year to that expected under the model parameterization as follows:

$$\Lambda_{Telemetry} = \sum_{i=1}^{Y} \frac{(v_i - n_i(1 - S_i))^2}{n_i(1 - S_i)},$$

where $S_i$ is again the annual survival rate in year $i$, $n_i$ is the number of collared animals alive at the beginning of year $i$, and $v_i$ is the number of collared adult moose that died in year $i$.

We then used a spectral projected gradient method using the "spg" function in the BB package in Program R [53] to numerically solve for the minimum chi-square estimate. This allowed us to directly estimate annual survival (i.e., $S_i$), initial cohort abundances in year 1 (i.e., $N_{11}$, $N_{12}$, $N_{13}$), and recruitment in subsequent years (i.e., $N_{21}$, $N_{31}$,. . .,$N_{Y1}$). All other female and male adult abundances were estimated based on the invariance property:

$$N_{i2} = (N_{i-1,1} \times 0.5 + N_{i-1,2}) \times S_i,$$

$$N_{i3} = (N_{i-1,1} \times 0.5 + N_{i-1,3}) \times S_i.$$

We calculated standard errors (SEs) for the minimum chi-square estimates using a numerical estimate of the inverse Hessian [48, 49, 54] using the "numDeriv" package in Program R [55]. Because reconstruction models consistently underestimate uncertainty [50], we inflated all standard errors by the goodness-of-fit scale parameter suggested by previous research [56]:

$$\sqrt{\frac{\chi_{df}^2}{df}},$$

where the $\chi_{df}^2$ statistic is based on the observed aerial survey data ($a_{ij}$) and their expected values under the reconstruction ($N_{ij}$). The degrees of freedom (df) are equal to $A{\times}Y{-}K$, where $K$ is the number of parameters estimated by the reconstruction. We then used these inflated standard errors to construct 90% confidence intervals for the model-derived estimates of annual population abundance and recruitment for moose in MN.

### Sensitivity analysis of reconstruction estimates

Given the rapid decline in animals seen during aerial surveys between 2009 and 2013 (64.8% in five years), we investigated the sensitivity of reconstructed population estimates during these years by incrementally increasing either adult survival or recruitment by 0.1%, while holding the other constant, until the population decline was reversed (i.e., population abundance in 2013 was within 10% of that in 2009).

### Population projection using reconstruction estimates

We projected our estimates of per-capita recruitment (number of calves that survived to one year per adult female alive during the previous year) and adult survival for an additional 10 years using the "forecast" package in Program R [57]. We then used the reconstructed estimates of calf, adult female, and adult male cohort abundance in 2020 as a starting point from which to predict cohort abundance from 2021 to 2030 using a stochastic version of the projection matrix approach described above [58].

## Results

### Survival estimates from collared moose

We did not detect a difference in overall survival in VNP by sex ($\chi_1^2 = 0.20$, $P = 0.70$). Adult annual survival estimates in years 2011 to 2017 for Voyageurs National Park ranged from 0.741 (95% CI 0.484–1.00) in 2015 to 1.00 in 2010 and 2013, with a mean annual survival of 0.893 (95% CI 0.833–0.958; Table 3). In 2010 and 2013, no collared moose mortalities occurred in VNP, precluding an estimate of variation in survival in those years. We did not detect a difference in overall survival in Grand Portage by sex ($\chi_1^2 = 0.60$, $P = 0.40$). Grand Portage adult annual survival in years 2010 to 2019 ranged from 0.591 (95% CI 0.417–0.837) in 2013 to 1.00 in 2012, with a mean annual survival of 0.833 (95% CI 0.794–0.874; Table 3). Because no collared moose mortalities occurred in Grand Portage Indian Reservation in 2012, we were precluded from estimating variation in survival.

### Population reconstruction of moose in MN

Using statistical population reconstruction with available aerial survey and telemetry data, we estimated fluctuations in adult survival, ranging from a maximum of 0.902 (SE = 0.043) in 2005 to a minimum of 0.690 (SE = 0.061) in 2019 (Fig 2). Per-capita recruitment (number of calves that survived to 1 year per adult female alive during the previous year) followed a similar

**Table 3. Adult moose survival estimates for Voyageurs National Park and Grand Portage Indian Reservation.** Estimates of annual survival and sex-ratios of collared adult moose in Voyageurs National Park and Grand Portage Indian Reservation, MN, USA, 2010–2021.

| Year | Voyageurs National Park | | | Grand Portage Indian Reservation | | |
|---|---|---|---|---|---|---|
| | Survival | 95% CI | F:M | Survival | 95% CI | F:M |
| 2010 | 1.000 | | 9:2 | 0.800 | 0.587–1.00 | 7:3 |
| 2011 | 0.895 | 0.767–1.000 | 13:6 | 0.667 | 0.466–0.953 | 12:3 |
| 2012 | 0.842 | 0.693–1.000 | 12:7 | 1.000 | | 12:0 |
| 2013 | 1.000 | | 10:4 | 0.591 | 0.417–0.837 | 20:3 |
| 2014 | 0.929 | 0.803–1.000 | 10:4 | 0.851 | 0.727–0.997 | 27:1 |
| 2015 | 0.741 | 0.484–1.000 | 9:2 | 0.781 | 0.658–0.928 | 35:3 |
| 2016 | 0.800 | 0.516–1.000 | 5:0 | 0.915 | 0.828–1.00 | 32:6 |
| 2017 | 0.750 | 0.426–1.000 | 4:0 | 0.866 | 0.752–0.998 | 27:8 |
| 2018 | | | | 0.851 | 0.726–0.998 | 23:8 |
| 2019 | | | | 0.931 | 0.843–1.00 | 20:9 |
| 2020 | | | | 0.778 | 0.659–0.918 | 34:10 |
| 2021 | | | | 0.887 | 0.806–0.977 | 40:18 |
| Overall | **0.893** | **0.833–0.958** | **72:25** | **0.833** | **0.794–0.874** | **289:72** |

cyclical pattern as adult survival, decreasing slightly from 0.386 (SE = 0.030) in 2005 to 0.303 (SE = 0.051) in 2019 (Fig 2). Winter moose abundance estimates showed a slow decline from an estimated 8,304 (90% CI = 7,797–8,788) animals in 2005 to 7,841 (90% CI = 6,702–8,933) in 2009 (Fig 3). This was followed by a sharp decline to 3,386 (90% CI = 2,681–4,243) animals in 2013, but remained steady afterwards to an estimated 3,163 (90% CI = 2,403–3,718) in 2020

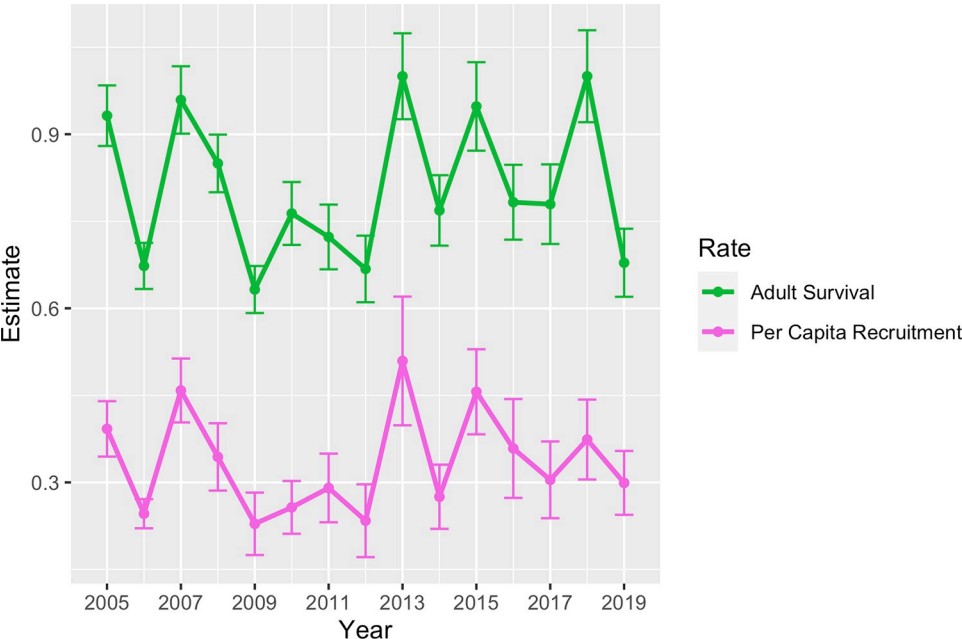

**Fig 2. Moose adult survival and fecundity estimates.** Estimated trends in annual survival (top) and per-capita recruitment (number of calves that survived to 1 year per adult female alive during the previous year) for moose in Minnesota (thick solid lines) between 2005 and 2019 based on statistical population reconstruction using integrated population models (IPMs), along with associated standard errors (error bars).

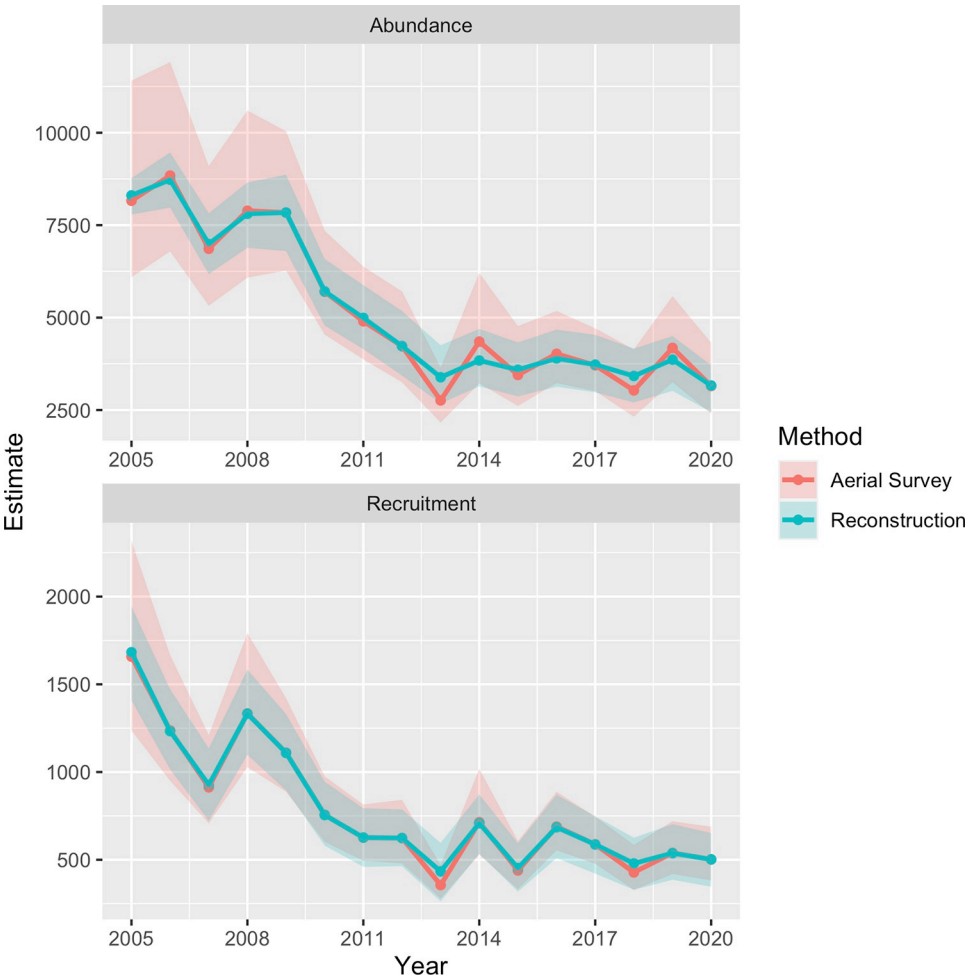

**Fig 3. Comparison of moose population estimates from reconstruction and aerial survey.** Estimated trends in abundance (top) and calf recruitment (bottom) into the winter population of moose in Minnesota (thick solid lines) between 2005 and 2020 based on statistical population reconstruction using integrated population models (IPMs), along with associated 90% confidence intervals (shaded regions).

(Fig 3). Annual recruitment followed a similar pattern and varied from a high of 1,683 (90% CI = 1,380–1,943) animals in 2005 to 502 (90% CI = 343–647) in 2020 (Fig 3).

## Sensitivity analysis of reconstruction estimates

Abundance estimates during the rapid decline from 2009 to 2013 were more sensitive to changes in adult survival than in recruitment. A 27.0% change in survival during the four years, while holding recruitment constant, resulted in a 2013 population abundance that was just 10% lower than that in 2009. To achieve a similar result while holding survival constant required an increase of 248.6% in recruitment during the four years.

## Population projection using reconstruction estimates

Stochastic projections using forecasted fecundity and survival estimates resulted in a slowly decreasing population from a high of 3,244 (90% CI = 2,936–3,461) in 2021 to a low of 2,680 (90% CI = 1,298–4,550) in 2030, and a corresponding annual growth rate of 0.984 (90% CI = 0.940–1.020; Fig 4).

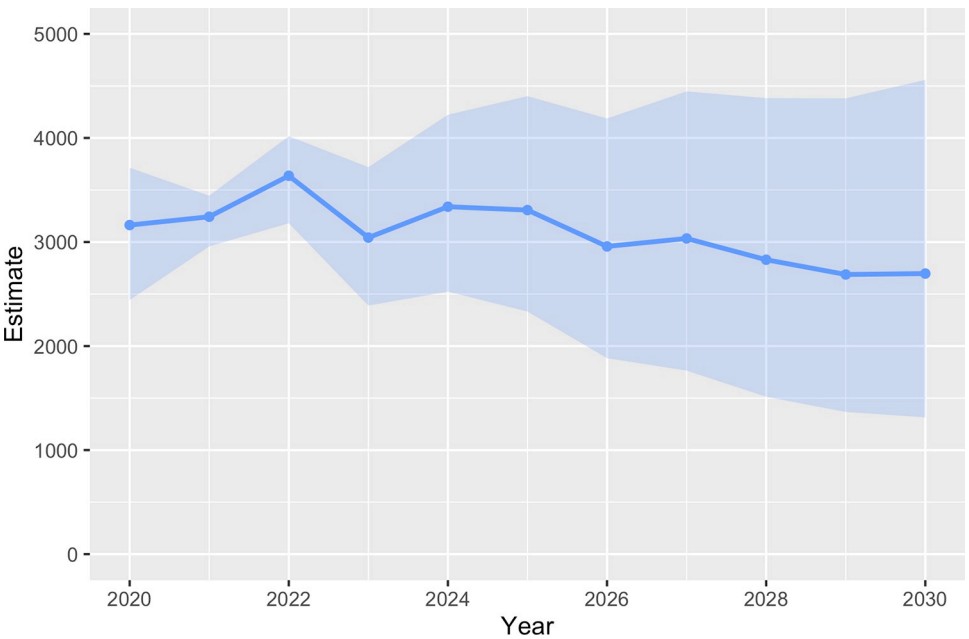

**Fig 4. Moose population projection.** Stochastic population projection of moose in Minnesota from 2020 to 2030 using forecasted estimates of annual survival and per-capita recruitment (number of calves that survived to 1 year per adult female alive during the previous year). Shaded regions represent 90% confidence intervals from 1,000 individual simulations.

## Discussion

Statistical population reconstruction was consistent with a substantial decline in the northeastern MN moose population between 2009 and 2013, as was indicated by the original aerial surveys conducted throughout the region. Reconstruction estimates indicated that during this time, the number of moose in primary moose range in MN decreased substantially from about 7,800 animals in 2009 to about 3,400 in 2013, corresponding to a >50% decline over just four years. Since 2013, however, the population largely stabilized and displayed an oscillatory pattern with a slight overall decrease of approximately 6.6% over the next seven years to an estimated 3,163 (90% CI = 2,403–3,718) animals in 2020. Stochastic projections using forecasted demographic rates indicated that this trend is likely to continue for the next 10 years to an estimated 2,680 (90% CI = 1,298–4,550) animals in 2030, yet the 90% CI of $\lambda$ included 1. This estimate closely matches simulated populations under a constant harvest of 150 adult males each year, but is less than populations under low harvest (40–80 adult males/yr; ~4,000 moose) [59].

Our results demonstrate the utility of using statistical population reconstruction to monitor moose population trends throughout northeastern MN and other parts of their North American range. When compared to estimates derived from aerial surveys, reconstruction estimates produced substantially narrower confidence intervals around similarly sized abundance estimates. For example, both aerial surveys and population reconstruction estimated similar abundances of 8,161 and 8,304 animals in 2005, respectively. However, the confidence interval around this reconstructed point estimate was approximately 20% of the confidence intervals around the aerial survey point estimate, a five-fold increase in precision. Although the increase in precision gained from reconstruction was substantially lower during many of the other years, reconstruction nonetheless provided a consistent improvement in precision when compared to estimates derived from aerial surveys (Fig 2). Moose sightability at the time of aerial

surveys, owing to individual moose behavior, habitat use, and weather, can contribute added variability affecting point estimates. Using statistical population reconstruction also eliminated the biologically unrealistic fluctuations in population abundance observed in the original aerial survey estimates. For example, aerial survey estimates indicated that moose abundance rebounded from 2,760 (2,160–3,650) animals in 2013 to 4,350 (3,220–6,210) in 2014, representing an increase of 57.1% in just one year. Given moose reproductive patterns, such a steep increase over such a short period of time is biologically impossible [52, 60]. Reconstruction estimates during the same time period, on the other hand, indicate an increase of only 13.4%, from 3,386 (90% CI = 2,681–4,243) to 3,840 (90% CI = 3,146–4,650), which is a reasonable increase given reported moose reproduction estimates [52, 60]. Conversely, the minimum modeled adult survival rate of 0.69 in 2019 may warrant caution, because such a low estimate is not biologically consistent with observed population trends and calf:adult female ratios [59].

An additional benefit of using statistical population reconstruction to monitor moose throughout the northeastern region is that it can retroactively provide abundance estimates during years when aerial surveys are not conducted. The COVID-19 pandemic prevented the statewide aerial survey for moose in 2021. After aerial survey and telemetry data are collected in subsequent years, statistical population reconstruction can be used to impute the missing number of calves, adult females, and adult males in 2021. Similarly, estimates of annual survival derived from telemetry studies often include years where no animals were monitored. In the present study, to our knowledge there were no published telemetry data collected in 2008 and 2009, precluding a direct estimate of survival during those years. However, with the use of statistical population reconstruction, we were able to estimate survival during those years.

Annual survival of moose in MN appears to follow a pattern of years of high survival followed by years of low survival (Fig 3). However, there was a consistent period of low survival between 2009 and 2013, corresponding to the observed and subsequently confirmed population decline of moose during this time [24]. Combined with the results of the sensitivity analysis, which indicated that population growth is more sensitive to fluctuations in adult survival than in per capita recruitment, these results suggest that the observed decline in population abundance was most likely caused by lower adult survival from 2009 to 2013. The two lowest collared moose survival rates measured on the Grand Portage Indian Reservation also occurred during this time period. Additionally, opportunistically collected free-ranging moose that were necropsied showed health and disease issues were common during this same period [39]. Subsequent research on cause-specific mortality of adult moose in northeastern MN further highlighted the significant effect of disease and parasites, such as winter tick and meningeal worm, on adult moose survival [24, 26, 39]. Additionally, wolf populations may have been subsidized by white-tailed deer in areas of moose range in MN, leading to declines in moose numbers via apparent competition and inverse-density-dependent predation [61]. Moose population dynamics, like those of many other large herbivores, are more impacted by variation in adult survival compared to juvenile survival [62, 63]. Adult survival typically varies little [63], but in populations exhibiting low and variable adult survival, populations decline [64]. Conversely, increases in adult survival can improve population performance [65, 66].

We believe our general approach was useful for a more comprehensive assessment of moose population dynamics of northeastern MN based on the integration of several different sources of information (i.e., aerial surveys and four separate telemetry studies). Future research should build upon this foundation to explore how the incorporation of other supporting data can improve reconstruction estimates and help to estimate additional model parameters not considered here. Data on annual pregnancy rates, calf survival, and twinning rates, for example, could be used to separate the effects of reproductive success from calf mortality, thereby allowing us to better identify the driving forces behind observed trends in annual recruitment.

Additional finer-scale studies, such as those ongoing at Grand Portage (S. A. Moore, unpublished data), that incorporate predator density, experimental manipulations of predator density, and effects of alternate (non-moose) prey of predators will be useful in teasing apart factors driving recruitment and mortality.

## Conclusion

Statistical population reconstructions confirmed that moose abundance in northeastern MN declined rapidly from 2009 to 2013 but has remained relatively stable during 2013–2020. Our results suggest that this decline was due primarily to low adult survival during those years. Our approach increased precision of population estimates gained from the state's annual aerial survey and can further be used to impute missing values when surveys cannot be conducted, such as occurred in 2021 due to the COVID-19 pandemic. Continued monitoring of vital rates of collared moose through the use of telemetry, such as that continuing to be undertaken by Grand Portage Band of Lake Superior Chippewa on Grand Portage Indian Reservation and in ceded territory in Superior National Forest, will aid in refining future estimates of population trends and projections and contribute to more precise knowledge of the population across time. As of publication, a moratorium on state-permitted collaring of moose is still in effect; this order does not restrict tribal activities (Executive Order 15–10, 28 Apr 2015). Without additional data streams to inform the aerial survey estimates, projections are less useful to managers of moose populations, especially when explicit mechanisms driving the trends are unknown.

## Acknowledgments

This material is based upon work supported in part by the Center for Applied Mathematics (CAM) Summer Research Program at the University of St. Thomas. We thank T. Arnold for inspiring this work, J. Fieberg for providing parameters from Lenarz et al. (2010), the MNDNR Wildlife Health Program, and dozens of volunteers and technicians for all moose capture and monitoring work. Thank you to T. Jung and an anonymous reviewer for comments that improved the clarity of the manuscript.

Federally recognized Indian tribes in northeastern Minnesota, USA, including the Grand Portage Band of Lake Superior Chippewa, Bois Forte Band of Chippewa, and Fond du Lac Band of Lake Superior Chippewa, proudly exercise their rights to food sovereignty through subsistence hunting and fishing. Moose are a primary subsistence food used by the Anishinaabeg (people) historically and presently. Management for and research on maintaining this moose population as a vital subsistence species thus sets the context for this paper examining the population trends of this culturally important resource.

## Author Contributions

**Conceptualization:** William J. Severud, Sergey S. Berg, Connor A. Ernst, Seth A. Moore.

**Data curation:** William J. Severud, Glenn D. DelGiudice, Seth A. Moore, Steve K. Windels, Ron A. Moen, Edmund J. Isaac, Tiffany M. Wolf.

**Formal analysis:** William J. Severud, Sergey S. Berg, Connor A. Ernst.

**Funding acquisition:** Sergey S. Berg, Glenn D. DelGiudice, Seth A. Moore, Steve K. Windels, Ron A. Moen, Tiffany M. Wolf.

**Investigation:** William J. Severud, Sergey S. Berg, Glenn D. DelGiudice, Edmund J. Isaac.

**Methodology:** William J. Severud, Sergey S. Berg, Connor A. Ernst, Ron A. Moen.

**Project administration:** William J. Severud, Sergey S. Berg.

**Software:** William J. Severud.

**Supervision:** William J. Severud, Sergey S. Berg.

**Validation:** Sergey S. Berg.

**Visualization:** William J. Severud, Sergey S. Berg.

**Writing – original draft:** William J. Severud, Sergey S. Berg, Connor A. Ernst.

**Writing – review & editing:** William J. Severud, Sergey S. Berg, Glenn D. DelGiudice, Seth A. Moore, Steve K. Windels, Ron A. Moen, Edmund J. Isaac, Tiffany M. Wolf.

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
