## [Decision Letter · Decision Letter 0]

25 Jul 2022

PONE-D-22-16773Statistical Population Reconstruction of Moose (Alces alces) in Northeastern Minnesota using Integrated Population ModelsPLOS ONE

Dear Dr. Severud,

Thank you for submitting your manuscript to PLOS ONE. After careful consideration, we feel that it has merit but does not fully meet PLOS ONE’s publication criteria as it currently stands. Therefore, we invite you to submit a revised version of the manuscript that addresses the points raised during the review process.

We received two sets of reviews. Both are very positive about the manuscript. However, the reviewers make many suggested revisions, which are mostly minor, but important for improving the manuscript.

We look forward to receiving your revised manuscript.

Kind regards,

Masami Fujiwara, PhD

Academic Editor

PLOS ONE

Journal Requirements:

"This material is based upon work supported in part by the Center for Applied Mathematics (CAM) Summer Research Program at the University of St. Thomas. We thank T. Arnold for inspiring this work, J. Fieberg for providing parameters from Lenarz et al. (2010), the MNDNR Wildlife Health Program, and dozens of volunteers and technicians for all moose capture and monitoring work. Collaring work was funded by Voyageurs National Park, a grant from the USGS-NPS Natural Resource Preservation Program, University of Minnesota-Duluth, U.S. Fish and Wildlife Service Tribal Wildlife Grant, U.S. Environmental Protection Agency Great Lakes Restoration Initiative, and the Bureau of Indian Affairs Endangered Species Program, Minnesota Zoo Ulysses S. Seal Conservation Fund, Indianapolis Zoo Conservation Fund."

"CAE was financially supported in part by the Center for Applied Mathematics Summer Research Program at the University of Saint Thomas (https://cas.stthomas.edu/departments/areas-of-study/mathematics/center-for-applied-mathematics/). Moose collaring work was funded by Voyageurs National Park (https://www.nps.gov/voya/index.htm; SKW and RAM), the USGS-NPS Natural Resource Preservation Program (PMIS 140435; SKW and RAM), University of Minnesota-Duluth (https://nrri.umn.edu/; SKW and RAM), U.S. Fish and Wildlife Service Tribal Wildlife Grant (https://www.fws.gov/service/tribal-wildlife-grants; SAM), U.S. Environmental Protection Agency Great Lakes Restoration Initiative (https://www.glri.us/node/443, SAM), and the Bureau of Indian Affairs Endangered Species Program (https://www.bia.gov/bia/ots/division-natural-resources/branch-fish-wildlife-recreation/endangered-species-program; SAM), Minnesota Zoo Ulysses S. Seal Conservation Fund (https://mnzoo.org/conservation/around-world/ulysses-s-seal-conservation-grant-program/; TMW), Indianapolis Zoo Conservation Fund (https://www.indianapoliszoo.com/conservation/field-support/; TMW). The funders had no role in study design, data collection and analysis, decision to publish, or preparation of the manuscript."

Reviewers' comments:

Reviewer's Responses to Questions

**Comments to the Author**

1. Is the manuscript technically sound, and do the data support the conclusions?

Reviewer #1: Yes

Reviewer #2: Yes

2. Has the statistical analysis been performed appropriately and rigorously? 

Reviewer #1: Yes

Reviewer #2: Yes

3. Have the authors made all data underlying the findings in their manuscript fully available?

Reviewer #1: Yes

Reviewer #2: Yes

4. Is the manuscript presented in an intelligible fashion and written in standard English?

Reviewer #1: Yes

Reviewer #2: Yes

5. Review Comments to the Author

Reviewer #1: I thought this was an interesting study about moose populations in NE Minnesota. It was well-written and easy to read. The statistical analysis appeared to be conducted appropriately and rigorously. I do not know if the authors made all data underlying the findings of their manuscript fully available, but I selected "yes" for lack of a appropriate category. I have only a few minor comments, clarifications, and edits listed below.

Additional Comments to Authors.

Line 112: Add “VNP” after Voyagers National Park?

Lines 113-114: “1992-1971”? Is this a typo? If not, seems weird to list the years going backwards.

Lines 122, 128: Change “This region” to “Our study area” to avoid ambiguity with “This”

Line 130: what does “relatively very few” mean? It could mean anything.

Line 138: “has” to “have”?

Line 193: “Previous reconstructions have then used an age-at-harvest matrix to represent each cohort”. I’m not clear what the age-at-harvest matrix looks like or how it’s implemented. The citations provided don’t appear to help understand this; they are focused on harvest reports. Also, finding the references for 31 and 32 was not trivial using a google search.

Line 205: your objective function only scores model predictions and does not account for model complexity. Won’t this by default always select the model with the most flexibility (i.e., the most complex model)?

Line 254: Shouldn’t the upper limit be 1 instead of 0.929 due to years 2010 and 2013 when no moose died? Or are you excluding these years because variability was 0?

Line 259: Same as previous comment. Shouldn’t 0.931 be 1.0 due to 2012?

Lines 303-308: Bias variance trade-off: Does your increase in precision affect accuracy of the estimator? Is the estimator unbiased?

Discussion: I would have liked to see a sentence or two about life-history strategies of moose in light of your survival and fecundity estimates. If moose have a slow-paced life-history strategy, it is not surprising that reduced adult survival really affected the population. Are these periods of reduced survival rates seen by adults that greatly affected the population (2009-2013) common in other animals with similar life-history strategies? How have other populations with similar population declines and life-history strategies responded? That is, is there hope that the NE MN moose population will recover to estimates in the 8-thousands? Or will they be in the 3-thousands until another episode of reduced adult survival drops abundance even lower?

Similarly, without knowing too much about moose life-history strategies, I was surprised there was no detectible difference in survival between sexes. Is this consistent with other moose studies? Is not being able to detect a difference due to lack of precision in the data/estimators? In the taxa I work with, females tend to have lower survival than males due to the cost of reproduction. In summary, why didn’t you find a difference in survival of the sexes? Was it because a difference doesn’t exist with moose, or is it because the data/esitimators were insufficiently precise to detect a difference if a difference existed?

Finally, there is some talk about cause-specific mortality regarding parasites and disease. However, it is not clear to me what the causes of mortality were between 2009-2013 that cut the population in half. Some more expert opinion/speculation of the causes of mortality and the relative effect of each on survival would be welcome in my opinion.

Reviewer #2: Dear Authors,

For me, this is an important study because 1) there is considerable concern in much of North America about local population declines of moose and your study is a key contribution that exemplifies the problem, and 2) we really need more case studies on the use of integrated population models to increase rigor and precision in wildlife management and conservation science by using all available data. Thank you for doing this work.

Overall, I had few issues with your manuscript, which is generally well written. Most were on issue of presentation and style, which I note below.

That said, my most substantive comments pertain to the population projection to 2030, which I suspect will be very important for local managers. Much more clarity on how these were conducted is needed, as are considerable cautions on the utility and ambiguity of the results. As is, I don't find this part of the manuscript useful, which is unfortunate.

I hope you find some of my comments helpful in revising your manuscript.

Sincerely,

Thomas Jung

Senior Wildlife Biologist, Government of Yukon

Adjunct Professor, University of Alberta

General Comments:

1) Mortality due to hunting of different sexes in the population projection to 2030 is not at all well described in the Methods or Results and it needs to be. I found it impossible to understand what the modeled projections pertain to – that is, were these for a hunted or unhunted populations, and if the former what levels of hunting for each sex. My concern is that the population projections are not going to be very useful for decision making without being much clearer on if and how hunting mortality was explicitly modeled.

2) Given that adult mortality appears to be driving the decline and population dynamics of the population, I would like to see substantial discussion of the causes of mortality in the Discussion, explicitly including hunting.

3) I would appreciate some discussion and cautions regarding the projected population estimates for 2030, given the wide CI and that the CI for lambda overlap 1.

4) Ensure the tense is consistently in the past. It is not in much of the Methods, for example (e.g., Lines 120, 128, 131, etc). A careful revision is required.

Detailed Comments:

Line 25: The paper would have wider appeal if you could expand this to more than just Minnesota. I believe that recent declines in moose abundance are happening broadly across North America?

Line 28: As per above, delete “across the state”

Line 36: Replace “ten” with “10”

Line 38. 0.689 is a really low annual survival estimate for an adult female ungulate.

Line 41: Rephrase without saying “leading us to conclude”

Lines 49-55: While I appreciate the interest for including this position statement, I think it would fit better in the main text (Introduction or Study Area) or Acknowledgements. The editors may be able to provide more specific guidance.

Line 60: Perhaps replace “parameters such as these” with “abudance and demographic parameters”

Line 69: Not only different studies, but likely more importantly is the ability of IPMs to incorporate disparate datasets from the same population (e.g., aerial surveys, radio-collared individuals, age-at-harvest, etc.).

Line 74: Replace “already” with “previously”

Lines 88-91: Simplify to: “In response to the rapid decline of moose in northeastern Minnesota, studies of adult and calf survival and cause specific mortality were initiated (Fig. 1).”

Line 92: Delete “state-of-the-art”

Line 98: Simplify to: “Specifically, we used statistical population reconstruction to estimate population abundance, recruitment, and estimate survival rates, using all available data.”

Line 104: Please note WHY you projected the estimates 10 years in the future (i.e., the management/conservation interests).

Line 108: Rephrase to: “Our study occurred in northeastern Minnesota, near the southern limit of the distributional range of moose”

Line 113: Please check “1992-1971”. Should this be “1971-1992”or other? Confusing.

Lines 113-118: This is confusing. I am unclear if there is a difference between state-licensed hunters and tribal subsistence harvest. In these years was all hunting closed, or just state-licensed hunters? Please rephrase carefully for clarity and concision. Moreover, these sentences should come closer to the end of the Study Area description.

Line 120-122: This sentence really should come after the first sentence in the first paragraph (i.e., Line 108).

Line 121: Delete: “between 47°06′N and 47°58′N latitude and 90°04′W and 92°17′W longitude”

Line 128: Define “primary” moose range please. I have no idea what that is.

Line 131: “MN” is not indicated as Minnesota earlier in the text. Please do so.

Line 138: Delete “has”

Line 139: When in winter? Provide month(s). This is relevant for classification methods if antlers were used to distinguish between males and females.

Line 146: Change “visually identified” to “classified”. Also, briefly discuss how whtis was done (e.g., body size, presence of vulva patch, antlers, etc.).

Line 148: Avoid use of “cows and bulls”. For an international journal use adult males and adult females. Make this change throughout the text, tables, and figures, where appropriate.

Lines 163-172: Simplify to: “We used annual adult survival rates from two previous studies. The first study used 150 adult moose (95 F/55 M) collared during 2002-2008 [11]; however, we only used survival rates that coincided with the aerial survey (2005–2007). We used pooled adult survival estimates because there was no difference in survival between males and females [11]. The second study was conducted from 2013-2016 and used 173 adult moose (123 F/50 M) [24]. Differences in survival between males and females were not reported, so we also used the pooled adult survival estimates from this study. Details of animal capture, handling, collaring, and monitoring can be found in the source publications [11,14,24,40].”

Line 173: Replace “2” with “two”

Line 193: Cite sources here please.

Line 197: Sure, but in Line 212 you assumed a 50:50 sex ratio of calves so you should note that here, rather than pooled in the model.

Line 199: You need to define an “adult”

Line 207: use “adult females” only.

Line 222: I think you mean “annual” not “yearly” survival?

Line 240: What were the increments?

Line 244: Replace “1” with “one”

Line 280: How was mortality by hunting used in the population projections, if at all? This is important given the varied hunting history of moose in this region. Are these estimates without hunting?

Line 283: That’s a wide confidence interval for 2030, with a lambda CI that ranges above and below 1. Just a comment.

Line 296: Okay. What does this mean? Are the projected estimates inclusive of harvest? This needs much better clarification as wildlife managers really need to know if and how the modeled projections include hunting and at what levels for each sex.

Line 308: Suggest change from “weather conditions” to “moose sightability”. This is because other factors such as individual behaviour or habitat use, as well as weather, can affect estimates. Moreover, aerial surveys should not be done in weather that results in substantially reduced sightability!

Line 344-357: I would suggest deleting this paragraph as it is quite tangential to the work.

Line 358: This is not an initial assessment. Suggest replacing “initial” with “refined” or “more comprehensive”

Line 377: Delete “such as that continuing to be undertaken by Grand Portage Band of Lake Superior Chippewa on Grand Portage Indian Reservation and in ceded territory in Superior National Forest,”

6. PLOS authors have the option to publish the peer review history of their article (what does this mean?). If published, this will include your full peer review and any attached files.

Reviewer #1: No

Reviewer #2: **Yes: **Thomas Jung

---

## [Author Response · Author response to Decision Letter 0]

8 Sep 2022

Thank you for your reviews. We have responded to each concern in the Response to Reviewers document.

---

## [Editor Report · Decision Letter 1]

12 Sep 2022

Statistical population reconstruction of moose (Alces alces) in northeastern Minnesota using integrated population models

PONE-D-22-16773R1

Dear Dr. Severud,

We’re pleased to inform you that your manuscript has been judged scientifically suitable for publication and will be formally accepted for publication once it meets all outstanding technical requirements.

Kind regards,

Masami Fujiwara, PhD

Academic Editor

PLOS ONE
---

## [Editor Report · Acceptance letter]

16 Sep 2022

PONE-D-22-16773R1 

Statistical population reconstruction of moose (*Alces alces*) in northeastern Minnesota using integrated population models 

Dear Dr. Severud:

I'm pleased to inform you that your manuscript has been deemed suitable for publication in PLOS ONE. Congratulations! Your manuscript is now with our production department. 

Kind regards, 

on behalf of

Dr. Masami Fujiwara 

Academic Editor

PLOS ONE